# Entomoculture: A Preliminary Techno-Economic Assessment

**DOI:** 10.3390/foods11193037

**Published:** 2022-09-30

**Authors:** Reina Ashizawa, Natalie Rubio, Sophia Letcher, Avery Parkinson, Victoria Dmitruczyk, David L. Kaplan

**Affiliations:** 1Department of Biology, Tufts University, Medford, MA 02155, USA; 2Department of Biomedical Engineering, Tufts University, Medford, MA 02155, USA; 3School of Life Sciences, McMaster University, Hamilton, ON L8S 4L8, Canada

**Keywords:** cultured meat, cell-based meat, technoeconomic assessment, insect cells

## Abstract

Cultured meat, or the practice of growing meat from cell culture, has been experiencing rapid advances in research and technology as the field of biotechnology attempts to answer the call to fight climate change and feed a growing global population. A major hurdle for cell-based meat products entering the market in the near-future is their price. The complex production facilities required to make such products will require advanced bioreactor systems, resources such as energy and water, and a skilled labor force, among other factors. The use of insect cells in this process is hypothesized to address some of these costs due to the characteristics that make them more resilient in cell culture when compared to traditional livestock-derived cells. To address the potential for cost savings by utilizing insect cells in the cultivation of protein-enriched foods, here we utilized a techno-economic assessment model. Three different insect cell lines were used in the model. The results indicate that insect cell lines offer potential to significantly reduce the cost per kilogram of cell cultivated meat, along with further opportunities to optimize production processes through technological advances and scaling.

## 1. Introduction

A recent report issued by the Intergovernmental Panel on Climate Change (IPCC) identifies human greenhouse gas (GHG) emissions as the unequivocal cause of global warming and outlines the alarming trajectory of the climate crisis if immediate action is not taken [1]. The report states that little can be done to prevent the 1.5 °C warming over the next two decades, however human emissions will be the determining factor in whether the warming stops there or continues to 4.4 °C by 2100 [1]. The former scenario in which warming stops at 1.5 °C requires global emissions to reach net zero by mid-century.

Current industrial meat production practices have been widely recognized as unsustainable [2,3,4,5]. In addition to issues of resource usage and pollution, ruminant livestock are estimated to contribute up to 37% of the anthropogenic methane emitted into the atmosphere [5]. With the IPCC’s new projections, the need for drastic change within our food system is incontrovertible.

### 1.1. Cost of Cellular Agriculture

Biotechnology has answered this call for change with the budding field of cultivated meat, which aims to generate sustainable meat alternatives that do not require traditional animal production methods. The field aims to adapt tissue engineering technologies, previously largely reserved for medical applications, to culture animal muscle and fat tissue. However, questions remain on whether cultivated meat products can compete economically with conventional meats, which is a common issue faced by any new technology as it develops and scales for commercial goals. A primary concern is the current cost of production, which would drive the market price higher than meats typically found on grocery store shelves. The first cell cultivated lab grown burger, achieved by a Dutch research group led by Dr. Mark Post in 2013, cost USD 325,000 to produce [6]. Cost of production in the following years still exceeds conventional meat prices—private company claims from recent years include USD 1080/kg of cultured beef (Upside Foods in 2018) to USD 1800/kg of cultured tuna (Finless Foods in 2019) [7,8]. As of 2021, however, some startup companies have reportedly been able to reduce costs by up to 99% [9].

In 2022, Upside Food claims that they have dramatically reduced cost of production with the scaling of their process, while the Israeli-based company Future Meat announced that their cultured chicken breast reached USD 1.70 and a per-kilogram price of USD 17 [10,11]. Eat Just debuted its cultured chicken at a restaurant in Singapore, which utilizes the meat in three small dishes for the price of USD 23 [12]. As more companies continue to scale up their production, the goal of reaching price parity with conventional meat has become more feasible in the past few years.

### 1.2. Techno-Economic Assessment

Techno-economic assessment (TEA) involves the modeling of an industrial process to understand the economics of a technology. Prospective TEA often involve the use of future scenario cases to best forecast the economic viability of an emerging technology. Three such analyses exist on cultured meat to date. A recent TEA on cultured meat carried out by David Humbird predicts cost of production to be USD 37/kg for a fed-batch reactor scenario and USD 51/kg for a perfusion reactor scenario [13].

In their 2021 report, CE Delft and the Good Food Institute (GFI) concluded that under an efficient medium use scenario—adding lower concentrations of ingredients such as glucose, amino acids, and recombinant proteins—with lower estimates for media ingredient prices, the price per kilogram of meat is projected to be approximately USD 149/kg (this study was conducted using proprietary information and is therefore not reproducible and not peer-reviewed) [14]. By altering factors such as cell density, production run time, cell volume, and recombinant protein costs, albeit rather optimistically, this future scenario price projection was reduced to USD 5.66/kg [14].

In 2020, researchers from the Spang Lab at the University of California, Davis published a preliminary techno-economic assessment of animal cell-based meat (ACBM) products from mammalian cells [15]. The assessment made price-per-kilogram projections for ACBM across four scenarios based on current production methods and prospective technological improvements [15]. The first assessment, based on current technologies and media costs from 2019, had a projected cost of USD 437,205 per kilogram. The second and third cost projections assumed scenarios in which current technical issues (e.g., media cost, glucose consumption rate, achievable cell concentration, doubling time) are addressed and costs reduced, yet prices still remained at USD 45,000 and USD 57,000 per kilogram, respectively. The study included the development of an open-source model to allow for further calculations of how different technological advances can impact the associated costs of production.

### 1.3. Insect Cells for Cellular Agriculture

Insects have long been explored as a sustainable future food source and can be highly nutritious [3,16]. While entomophagy remains uncommon in many in Western cultures, insects have been a staple in diets for centuries, and are more frequently consumed in certain parts of Asia, Africa, Australia, and South America [17].

While consumer perceptions are indeed a concern for both cellular agriculture and entomophagy, insect cells offer a compelling alternative to a few of the issues facing cultivated meat production costs. The use of insect cells for cellular agriculture presents interesting advantages over mammalian cells. Insect cells are more resilient to environmental factors during growth and require fewer resources (e.g., carbon dioxide, growth factors) than vertebrate cell cultures [18]. A major advantage of insect cells for cellular agriculture (i.e., entomoculture) is their adaptability to serum-free media, which is vital for cultured meat production as this can decrease media cost and variability as well as address ethical concerns [19].

Insect cell lines like Sf-9 (*Spodoptera frugiperda*) and S2 (*Drosophila melanogaster*) have already been utilized in biotechnology, particularly within the field of recombinant protein production [18]. This field has established a foundation for insect cell production for food applications, with the development of technologies for high density culture such as improved bioreactors and optimal media formulations [20,21]. All the above issues point to lower potential costs for the production of cultivated meat through the use of insect cells when compared to cells derived from livestock animals.

In the present work, the UC Davis ACBM cost calculator model was tailored to insect cell culture to estimate input requirements and costs of insect cell-based meat production. This report aims to dissect each outcome produced by the model to understand the differences in costs when producing mammalian and insect cultivated meat.

## 2. Materials and Methods

### 2.1. ACBM Cost Model

The open-source ACMB cost model developed by Risner et al. was used to carry out the techno-economic assessment of insect cell-based meat [15]. The cost model should be interpreted as a preliminary TEA, as certain processes such as seed train, scaffolding, microcarriers, and bioreactor cleaning are omitted. Downstream and post processing are also not considered within the original or present models, so the hypothetical product systems are assumed to yield products similar to minced meat. While these are certainly important factors for consideration in the cost estimation of cultured meat, holding included or excluded processes in the model constant between studies allows for comparison between the two cell types, which is the objective of the present study.

Three insect cell lines were the focus of this study—Sf-9, Hi-Five, and S2—chosen for their common use in the pharmaceutical industry and robust data availability. Both originating from Lepidopteran ovarian cells, the Sf-9 and Hi-Five cell lines had similar characteristics and were therefore analyzed together. S2 cells, which were established from *Drosophila melanogaster* embryonic tissue and are the most used *Drosophila* cell line, were analyzed independently due to their unique characteristics. Fourteen variables included in the code for mammalian cell culture were changed to reflect values for insect cell culture. These parameters are average single cell volume, incubation temperature, specific heat of meat, doubling time, achievable cell concentration, oxygen consumption rate, glucose consumption rate, glucose concentration in basal media, maturation time, basal media cost, and supplemental media ingredient concentrations (Table 1). These variables were either averages or best representative values determined through a review of the literature (see Appendix A
Table A1 and Appendix B
Table A8). Parameters generalizable to cell culture process were held constant (see Appendix A
Table A3). Basal media cost included for insect cells in Table 1 represented cost of complete media, whereas mammalian media included growth factors and other supplemental ingredients separately.

### 2.2. Sensitivity Analysis

The SALib Python package was utilized to perform Sobol Sensitivity Analysis (SSA) on 11 variables within the model, chosen because their values changed based on the type of cell culture being assessed. SSA was used to determine contribution of these variables to the variance of the output by evaluating first-order and total-effect indices. Variables found to have greater influence over the model outcomes were chosen for subsequent cost-minimization scenarios.

### 2.3. Media Cost Estimates

Media cost and other media-associated parameters were based on two types of insect cell media: Yeastolate-PRimatone (YPR) for the High Five/Sf-9 model and Schneider’s Drosophila Media for the Drosophila model [22]. The cost of producing the media in 20,000 L batches was determined using list pricing of each of the constituent components. Cost breakdown of basal medium used in these formulations was performed to determine the cost of raw ingredients. In doing so, cost of media associated with profit margins of suppliers was eliminated to match the assumption that media would be formulated by the production facility. See Table A4, Table A5, Table A6 and Table A7 in Appendix A for a breakdown of media costs.

### 2.4. Scenario Models

Cost-minimization scenarios identified key parameters in the model that could be leveraged to reduce cost of insect cell-based meat. Each scenario addressed a single parameter and offered a technical solution along with a new value for the said parameter based on literature review.

Scenario B required edits to be made within the code of the model beyond changing the parameter values. For this scenario it was proposed to use byproduct accumulation as a measure for media turnover rather than glucose consumption. The code written to calculate the rates of glucose consumption in the growth and maturation phases were instead used to determine rates of lactate production. For this scenario, pure glucose (rather than bulk media) was added as needed based on consumption rates. The calculation to determine the number of media changes (see parameter MediaChargeBatch in Table A2 in Appendix A) was edited to divide the total concentration of lactate produced by the cells by the concentration of lactate inhibitory for insect cells. It was found that levels of lactate rarely exceeded toxic concentrations, so a minimum value of 1 was assigned to the MediaChargeBatch parameter, as a smaller number would suggest that volumes of media less than 20,000 L were added to the bioreactor.

## 3. Results and Discussion

The cost of production of insect cell-based meat on a per kilogram basis can be seen broken down by input type in Table 2. Media cost was found to contribute to 99% of the overall cost, reiterating findings across all published cultured meat TEAs that indicate media as a major cost driver regardless of cell line [13,14,15,23]. Major differences between mammalian and insect cell line product systems were observed and discussed in detail in the subsequent sections.

### 3.1. Understanding Model Outcomes: Bioreactor Outcomes

Minimizing costs of bio-equipment and media are of top priority when looking for ways to bring cultivated meat toward a cost-competitive basis with traditional meat. To analyze these costs, it was first necessary to determine the number of the bioreactors needed to meet the annual production target. The number of batches able to be produced by a single bioreactor per year was based on total batch time, or the sum of cell growth time and maturation time. Batch time was generally reduced in insect cells by 3–4 days due to the more rapid population doubling rates for insect cells in comparison to mammalian cells. Cell mass per batch factors in the bioreactor working volume, achievable cell concentration, cell density, and cell volume. The mammalian model used a eukaryotic muscle cell density of 1060 kg m^−3^ reported by the Good Food Institute and a standard 20,000 L bioreactor volume, so these variables were kept constant in the modified insect model [23].

Due to lack of data on insect cell volumes, reported insect cell diameters were instead used to determine cell volumes with the assumption that cells were spherical, and the values for High-Five and Sf-9 cells were found to be about half of the mammalian cell size used in the original model (see Appendix A). Achievable cell concentration was determined from the literature to be about twice as high for insect cells compared to mammalian cells and has potential to increase with technological advances [24,25,26]. Cell mass per batch and batches per year were then used to calculate the total cell mass produced per year by one bioreactor. While the insect cells’ smaller cell volume was found to drive the value of this outcome down, their higher achievable cell density led to an overall increase in total cell mass produced.

The number of bioreactors needed in the facility was determined by dividing the desired mass of meat to be produced each year by the total annual production outcome. The desired mass of meat was 121,000,000 kg in both the animal and insect cell models, representing 1% of the current US beef market [15]. This value was then multiplied by batches per year from one bioreactor to get the total number of batches produced annually by the production facility. The cost of a single bioreactor in the insect model was calculated using the same assumptions as the original model: a USD 50,000 m^−3^ unit cost, 0.6 common scaling factor, and 1.29 adjusted value factor to account for inflation. This bioreactor cost was then multiplied by the number of bioreactors in the plant and a Lang factor of 2 which accounts for installation costs to arrive at a total cost for bio-equipment.

Fixed manufacturing cost represents the minimum capital expenditures needed to produce the desired quantity of meat and is calculated using the total bio-equipment costs and a fixed manufacturing cost factor of 0.15. While the various factors and unit costs used in these calculations were held constant between the mammalian and insect cell models, insect cell-based meat production required slightly fewer bioreactors because of the larger yield of cultivated meat per bioreactor, which resulted in decreased manufacturing costs (Table 3).

### 3.2. Understanding Model Outcomes: Media

Annual media cost is determined by the cost and annual volume of media needed. The original mammalian cell model determines media cost by using concentrations and pricing of growth factors and other components, then adding this supplementary cost to the cost of basal medium. Insect cells can grow without most of the components included in this model, therefore the insect cell model instead uses the predetermined cost of complete insect media.

Annual volume of media utilized in the bioreactor system is dependent on cellular metabolism. This model uses glucose consumption rate to approximate media requirements, assuming media must be replenished whenever glucose in the basal media is depleted. The total glucose consumed per batch is a sum of the glucose consumed in the growth and maturation phases. The glucose consumed in the growth phase is impacted by doubling time, glucose concentration in the basal media, and achievable cell concentration, while glucose consumed in the maturation phase is impacted by maturation time, achievable cell concentration, bioreactor working volume, and glucose consumption rate.

Due to shorter growth and maturation times, as well as slower glucose consumption rates as reported in the literature, total glucose consumed by insect cells is significantly decreased compared to mammalian cells. This also means that insect cells require fewer media changes and thus greatly decreases annual media cost for the production facility.

Like glucose consumption, oxygen consumption is broken down into consumption in the growth and maturation phases and calculated almost identically. Despite the reported oxygen consumption rates of insect cells generally being higher than that of mammalian cells, oxygen consumed per batch and annual oxygen consumption are lower in insect cells due to their faster growth rates (Table 4).

### 3.3. Understanding Model Outcomes: Utility

Variable operating expenses of cell-based meat production facilities include utilities, which in this model account for electricity and water. Total electricity used in a production facility was assumed to be the sum of energy needed to cool the bioreactors, heat the media, and cool the final meat product. Two important variables included in these energy calculations were the incubation temperature and specific heat of meat. Since insect cells are incubated around 27–28 °C, less energy is required to heat the media. The specific heat of insect meat was also lower than the specific heat of beef (see Appendix B), the value used for the mammalian cell model. While the number of bioreactors was factored into two of the three energy calculations, the need for more bioreactors in the insect cell model was outweighed by the energy saved by these factors as well as the lower media requirement, which resulted in a decreased energy cost within the insect cell model.

Process water required and wastewater produced was estimated using the annual volume of media, assuming that media would be produced onsite. Annual water cost was thus equal to the sum of total process water and wastewater costs. As mentioned above, annual volume of media outcome was found to be smaller in the insect cell model and thus annual water cost was reduced as a result (Table 5).

### 3.4. Understanding Model Outcomes: Labor

Labor related costs were also factored into the variable operating expenses of a cultured meat plant. The amount of manpower required was based on the number of bioreactors needed per year. Annual labor cost is calculated using this value, the average hourly rate of a meat packer, annual operation time, and a labor cost correction factor (Table 6). The insect model assumed the same wages, operation time, and cost correction factor, therefore the slight decrease in labor cost compared to the baseline mammalian scenario can be attributed to the need for fewer bioreactors for the insect facilities.

### 3.5. Understanding Model Outcomes: Financing

The original model uses several standard financial calculations for equity and debt, which remained unchanged for the modified insect model. Total equity and debt costs simply account for the cost of bioreactors multiplied by the given equity and debt ratios. Using these values and the calculated capital and debt recovery factors, annual equity recovery and annual debt payment can be found. Summed together, these values represent the total annual payment and can then be used to determine capital expenditure. Minimum annual operating cost is the sum of fixed manufacturing costs, annual media costs, annual oxygen costs, electric costs, annual labor costs, and annual water costs. After a reduction in many of these costs with the use of insect cells as outlined above, this overall cost was reduced up to 100-fold. Annual operating cost is divided by desired mass of meat to determine the minimum amount of meat produced to meet expenditures. Minimum annual capital and operating expenditure includes the operating costs as well as the bio-equipment total cost over the economic lifespan of the production facility, which is assumed to be 20 years in both models. This total cost is then divided by the desired mass of meat to finally determine the price of cell-based meat per kilogram (Table 7).

Again, due to reductions in media, oxygen, and utility costs, cost per kilogram of insect cell-based meat is significantly lower than the cost determined by the original mammalian cell model (Table 6). As already demonstrated in the original study, technological advancements have the potential to greatly reduce this base price to one that is cost-competitive with traditionally farmed meat.

### 3.6. Sensitivity Analysis

Sobol Sensitivity Analysis was performed on the 11 variables that had values changed to specifically represent insect cell characteristics for the purposes of our model (Table 8). Larger first order and total values indicated that the variable had a larger impact on the results of the model. This sensitivity analysis was carried out to determine areas in which the production system could be further improved to minimize cost.

### 3.7. Proposed Scenarios to Reduce Per Kilogram Cost of Insect Cell Cultured Meat

Using the results from the sensitivity analysis, the five most impactful variables were used to project different scenarios in which changes or technological improvements could be implemented in future insect cell-based meat facilities, as described below. The effects of these scenarios on cost of meat per kilogram are summarized in Figure 1.

#### 3.7.1. Scenario A: Larger Cell Size Increases Cell Mass Produced Per Batch

Average single cell volume was found to have the most significant effect on the final cost of meat per kilogram. Cell volume is factored in with single cell density, cell concentration, and bioreactor volume to determine the achievable cell mass per batch. The mass per batch parameter is then used to calculate the number of bioreactors needed and annual batches produced, with a smaller achievable cell mass resulting in a need for more bioreactors and more batches. These parameters impact all further cost calculations including those for media, oxygen, electricity, water, manufacturing, and labor. Thus, a smaller cell size drives up costs in all areas of production.

The first method for cost reduction of insect cell-based meat that we propose is the utilization of a larger cell type. The primary cell lines included in our literature review were Sf-9 (*Spodoptera frugiperda*), High-Five (*Trichoplusia ni*), and S2 (*Drosophila melanogaster*). S2 cells are the smallest of the three types, with their average volume (5.73 × 10^−16^ m^3^, *s* = 1.75 × 10^−16^ m^3^) [27,28] coming in an order of magnitude smaller than the other two lines. High-Five cells (2.02 × 10^−15^ m^3^, *s* = 3.56 × 10^−16^ m^3^) [29,30] are generally slightly larger than Sf-9 cells (2.30 × 10^−15^ m^3^, *s* = 7.78 × 10^−16^ m^3^) (Table 9), making High-Fives the optimal cell type out of the three in terms of size.

Larger yet are the insect cells derived from ovarian tissue of the moth species *Antheraea eucalypti*, which were reported to have cell volumes significantly greater (1.41 × 10^−14^ m^3^) than the average sizes later determined for High-Five cells [31]. These cells were used by Thomas Grace in 1962 to establish the first continuous insect cell line and appear to still be in use today [32]. Although larger cell size may be associated with higher nutrient consumption rates, other parameters were held constant for sake of simplicity in this scenario. Using the dimensions of the *Antheraea* clone “AeC6” included in Grace’s initial 1968 report to approximate the single cell volume of a larger cell line, the price of insect cell-based meat was projected to be USD 797.66/kg.

#### 3.7.2. Scenario B: Different Media Consumption Measurements May Be Used to Decrease Turnover Rates

Media contributes a significant cost to the production of cultured meat, but this cost may be reduced by minimizing the number of times the media must be replaced and thus the total volume of media required. The original model uses cellular metabolism of glucose to determine media replacements required per batch. We hypothesize that rather than turning over the media each time glucose is depleted, production facilities may supplement glucose separately from the bulk media, according to metabolic demand. For this scenario, we instead base media replacement requirements on the accumulation of lactate, which is inhibitory to insect cells at concentrations exceeding 12.5 mM [30]. We hypothesize that this would reduce the cost-contribution of media, energy, and water to effectively reduce the price of cell cultivated meat.

Glucose was made a supplement in the model, with a projected cost of USD 0.26/kg at large-scale, and a concentration of 0.9 g/L, which was determined by previous modeling results [13]. Lactate production rate of insect cells was found in the Neermann and Wagner study on Sf-9 cell metabolism [33]. Calculations for lactate accumulation were similar to those used for glucose consumption in the growth and maturation stages. As lactate accumulates at such a slow rate, it never exceeded the toxic level and therefore media only had to be turned over once (at the start of the batch). Seeing this was the case, we decided to try basing our model off ammonia accumulation as well. In keeping with the lactate calculations, ammonia production rates of Sf-9 cells were used. The upper range given by the 2007 Drugmand review was chosen to give a more conservative estimate [30]. The ammonia-based model results in a price approximately twice as high as the lactate-based scenario.

Running the model with both an increased cell size and a new basis for media turnover, the price of insect cell-based meat was reduced to USD 126.96/kg in the lactate-based scenario.

#### 3.7.3. Scenario C: Base Media Formulation and Supplementation May Be Altered for Cost Minimization

Media cost is a widely recognized driver of cell-based meat production cost [23]. Our base model scenario assumes the cost of Yeastolate-Primatone (YPR) medium, a serum-free insect cell culture medium developed by Ikonomou et al. in 2001 [22]. YPR cost was originally estimated to be USD 28.88/L (see Appendix A) based on the assumption that the medium was formulated in-house with IPL-41 as the basal medium and other ingredients sourced through bulk-pricing to minimize expense.

Numerous media-cost-reduction scenarios have been previously identified. One such way is to alter the amino acid composition in basal media by replacing the IPL-41 formulation with a defined basal media composition containing decreased amino acid concentrations. Previous studies on IPL-41 have found that only 26% of amino acids are utilized, and formulations with reduced amino acid concentrations did not impede cell growth [34]. Our calculations found that this strategy effectively reduces the in-house basal media cost from USD 1.34/L to USD 0.36/L (see Appendix A).

YPR uses the hydrolysates yeastolate ultrafiltrate and Primatone RL as serum substitutes to avoid the quality and ethical concerns that come with the use of animal serums. Hydrolysates offer similar medium supplementation of oligopeptides, amino acids, polysaccharides, and vitamins necessary for successful cell proliferation, however previous media cost minimization studies have found that rather simple replacements can be made in order to decrease cost [22]. Yeastolate ultrafiltrate (USD 1970/kg) can be replaced by yeast extract, another insect cell culture supplement that is offered at a significantly reduced cost (USD 5/kg). Primatone RL is not only a somewhat costly ingredient but also originates from animal tissue. By substituting Primatone with a soy hydrolysate such as HySoy, the cost of this hydrolysate component could be brought down from USD 620/kg to USD 2/kg while also making the media “animal-component-free”.

These changes to the media can bring the price of YPR from USD 28.88/L down to a mere USD 1.70/L. When running the model with this media price, the cost of insect cell-based meat comes down to USD 10.49/kg.

#### 3.7.4. Scenario D: Insect Growth Factors May Increase Achievable Cell Concentration, Thus Increasing Cell Mass Produced per Batch

Another foreseeable technological improvement to decrease insect cell-based meat cost is the achievement of higher cell densities in culture. Higher achievable cell concentration can drive down cost by increasing the mass of meat produced by production plants each year. One factor known to impact cell concentration is medium nutrient composition. Oftentimes, cell cultures will be supplemented with fetal bovine serum (FBS), but this is a byproduct of the meat industry, while in contrast, this model assumes the use of recombinant growth factors as animal serum alternatives. Cell cultures for cultivated meat can instead utilize these animal-free growth factors to increase proliferation, but as seen in the original model, these impose some of the most significant costs to production. As emphasized earlier in the text, insect cells can grow in the absence of these costly growth factors included in the original mammalian cell-based model, such as transforming growth factor beta and fibroblast growth factor 2. As increased cell concentration was found to be a considerable cost lever, other growth factors specific to insect cells have been identified and may be added to promote growth while not significantly contributing to cost of production.

One such growth factor is *Bombyx mori* paralytic peptide, an insect-derived polypeptide that increased cell proliferation by up to two-fold when added to culture media. Another lesser-studied insect growth factor is growth-blocking peptide, which was shown in a 1998 study by Hayakawa and Ohnishi to increase growth at low concentrations. More recent studies have begun to look at the polypeptide imaginal disc growth factor-2 (IDGF-2) that originates from *Drosophila* species but promoted lepidopteran cell growth [35,36,37]. In their 2006 study, Zhang et al. found that at concentrations greater than 0.2 nM, the growth factor increased cell concentration by up to 29% [38]. Despite all three of these growth factors showing promise for increasing achievable cell concentration, we decided to include IDGF-2 in our model for Scenario 3 due to the more recent focus on its applications in insect cell culture.

Based on the assumption that the addition of this growth factor at 40 ng/mL can increase cell concentration by 29%, our new achievable cell concentration increased to 2.7 × 10^7^ cells/mL. Since it is not commercially available, the price of IDGF-2 was assumed to be equal to the cost of FGF-2 included in the original model (USD 2,005,000/g). Despite its assumed high price, IDGF-2 supplementation is still projected to decrease the price of insect cell-based meat to USD 7.78.

## 4. Conclusions

Due to reductions in media, oxygen, and utility costs, the baseline cost-per-kilogram of insect cell-based meat—USD 4193 for Lepidopteran-based and USD 6426 for Drosophila-based—is significantly lower than the cost determined by the original mammalian cell model, determined by Risner et al. to be USD 437,205 [15]. While both insect cell lines generated lower cost projections, Lepidoteran cells were found to reduce cost most dramatically (Table 6). This outcome was likely due to their larger cell size, shorter doubling time, and lower media cost, three parameters in the model that were found to have significant impact on cost (Table 7).

Companies in the cellular agriculture space may be more compelled to consider insect-cell lines for product development or take note of insect cells’ attractive traits for possible areas of optimization in their own cell lines. The present study also highlights media optimization as a hot spot for future cost-reduction strategies. Such strategies include increasing media use efficiency, cell-line engineering for metabolic efficiency, and media recycling. Ingredient sourcing is another lever with high potential for cost-reductions, as explored in Scenario C. With its traditional applications in pharmaceuticals and biomedical research, cell culture media has not been produced at the grade or scale of industrial food production. This offers a substantial opportunity for cost-savings as cultured meat companies scale commercially.

Through technological advancement, cell line optimization, and economies of scale, insect cells have been modeled here to offer attractive qualities for a cultured meat product able to undercut cost-per-kilogram of conventional beef, currently valued at USD 26.38 in the USA [39]. As a preliminary TEA, the present study is limited in its ability to accurately forecast cost of production for insect cell-based meat. Further research into this topic may explore costs associated with scaffolding or downstream processing associated with achieving a more conventional-meat-like product (i.e., 3-D structure, texture, flavoring, etc.). Media should be further The present TEA model can confirm the hypothesis that cultivated meat can achieve price parity with traditional meats more readily using insect cells compared to mammalian cells.

## Figures and Tables

**Figure 1 foods-11-03037-f001:**
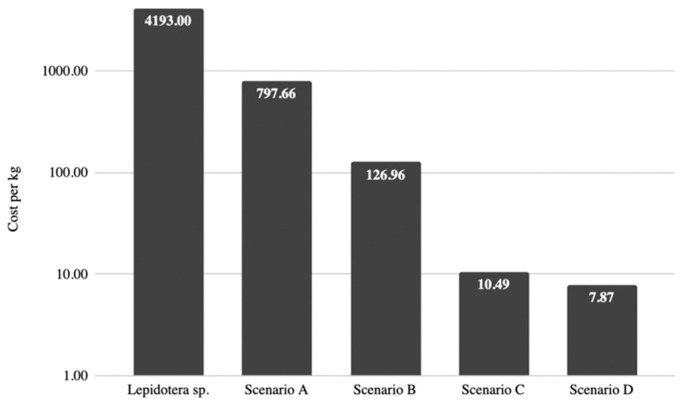
Summary of calculated costs of meat per kilogram according to model for each proposed cost reduction scenario.

**Table 1 foods-11-03037-t001:** Variables changed within code based on cell type with variable description, units, mammalian values from Scenario 1, and baseline insect values. See Appendix A and Appendix B for insect value calculations and references [15].

Variable Name	Description	Units	Mammalian [15]	Sf-9/Hi-Five	S2
desired_Temp	Cell incubation temperature	C	37	27	28
aveCellVol	Average volume of single cell	m^3^/cell	5.00 × 10^−15^	2.16 × 10^−15^	5.73 × 10^−16^
Ug	Glucose consumption rate per cell	mol/h·cell	4.13 × 10^−13^	9.61 × 10^−14^	1.51 × 10^−14^
GConInBM	Glucose concentration in basal media	mol/L	1.78 × 10^−2^	5.55 × 10^−2^	1.11 × 10^−2^
oxygen_consump	Oxygen consumption rate per cell	mol/h·cell	1.80 × 10^−14^	3.07 × 10^−13^	1.12 × 10^−14^
MatTime	Time until cell maturation	h	240	168	168
ACC	Highest achievable cell concentration in culture	cells/mL	1.00 × 10^7^	2.00 × 10^7^	3.01 × 10^7^
d	H per population doubling	h	24	22.72	38.50
BaseMedia_cost	Cost of culture media	USD/L	3.12	28.88	13.65
ACBM_spec_heat	Specific heat of meat product	kWh/kg·C	6.22 × 10^−4^	9.43 × 10^−5^	9.26 × 10^−5^

**Table 2 foods-11-03037-t002:** Breakdown of cost of production per kilogram of insect cell-based meat.

	Hi-Five/Sf-9	S2
Media Cost	USD 4186.78	USD 6362.97
Water Cost	USD 0.25	USD 0.80
Electricity Cost	USD 0.74	USD 1.96
Oxygen Cost	USD 0.15	USD 0.38
Manufacturing Cost	USD 4.55	USD 15.21
Labor Cost	USD 11.78	USD 39.33

**Table 3 foods-11-03037-t003:** Outcome values relating to bioreactors produced by the model for four mammalian scenarios proposed in the Risner et al. paper compared to baseline outcomes for Hi-Five/Sf-9 and S2 insect cells [15].

		Mammalian [15]				Insect	
	Units	Scenario 1	Scenario 2	Scenario 3	Scenario 4	Hi-Five/Sf-9	S2
Batches per bioreactor per year		22	34	34	114	28	21
Cell mass per batch	kg	1.06 × 10^3^	1.01 × 10^4^	1.01 × 10^4^	2.12 × 10^4^	9.16 × 10^2^	3.66 × 10^2^
Cell mass produced per bioreactor per year (kg)	kg	2.33 × 10^4^	3.42 × 10^5^	3.42 × 10^5^	2.42 × 10^6^	2.56 × 10^4^	7.68 × 10^3^
No. bioreactors per year		5.19 × 10^3^	3.54 × 10^2^	3.54 × 10^2^	5.10 × 10^1^	4.72 × 10^3^	1.58 × 10^4^
Total no. batches produced annually		1.14 × 10^5^	1.20 × 10^4^	1.20 × 10^4^	5.81 × 10^3^	1.32 × 10^5^	3.31 × 10^5^
Total cost of bioreactors	USD	4.04 × 10^9^	2.76 × 10^8^	2.76 × 10^8^	3.97 × 10^7^	3.67 × 10^9^	1.23 × 10^10^
Fixed manufacturing cost	USD	6.06 × 10^8^	4.13 × 10^7^	4.13 × 10^7^	5.95 × 10^6^	5.51 × 10^8^	1.84 × 10^9^

**Table 4 foods-11-03037-t004:** Outcome values relating to media produced by the model for four mammalian scenarios proposed in the Risner et al. paper compared to outcomes for Hi-Five/Sf-9 and S2 insect cells [15].

		Mammalian [15]				Insect	
	Units	Scenario 1	Scenario 2	Scenario 3	Scenario 4	Hi-Five/Sf-9	S2
Conc. glucose in bioreactor	mol	356	534	534	712	1110	222
Total glucose consumed per batch	mol	2.19 × 10^4^	6.79 × 10^4^	6.79 × 10^4^	5.34 × 10^3^	7.37 × 10^3^	1.89 × 10^3^
No. media changes per batch		61	127	127	8	7	9
Volume media used per batch	L	1.23 × 10^6^	2.54 × 10^6^	2.54 × 10^6^	1.50 × 10^5^	1.33 × 10^5^	1.70 × 10^5^
Volume media used annually	L	1.40 × 10^11^	3.06 × 10^10^	3.0 × 10^10^	8.72 × 10^8^	1.75 × 10^10^	5.64 × 10^10^
Annual media cost for facility	USD	5.29 × 10^13^	6.93 × 10^12^	5.40 × 10^12^	2.09 × 10^8^	5.05 × 10^11^	7.70 × 10^11^
Oxygen consumption per batch	mol	7.70 × 10^5^	1.60 × 10^6^	1.60 × 10^6^	9.61 × 10^4^	1.06 × 10^5^	1.08 × 10^5^
Annual oxygen consumption	g	2.81 × 10^6^	6.15 × 10^5^	6.15 × 10^5^	1.79 × 10^4^	4.50 × 10^5^	1.14 × 10^6^
Annual oxygen cost	USD	1.12 × 10^8^	2.46 × 10^7^	2.46 × 10^7^	7.15 × 10^5^	1.80 × 10^7^	4.57 × 10^7^

**Table 5 foods-11-03037-t005:** Outcome values relating to utilities produced by the model for four mammalian scenarios proposed in the Risner et al. paper compared to outcomes for Hi-Five/Sf-9 and S2 insect cells [15].

		Mammalian [15]				Insect	
	Units	Scenario 1	Scenario 2	Scenario 3	Scenario 4	Hi-Five/Sf-9	S2
Electricity cooling bioreactor	kWh	1.14 × 10^10^	2.50 × 10^9^	2.50 × 10^9^	7.26 × 10^7^	1.83 × 10^9^	4.64 × 10^9^
Electricity heating media	kWh	3.82 × 10^9^	8.33 × 10^8^	8.33 × 10^8^	2.37 × 10^7^	1.96 × 10^8^	7.22 × 10^8^
Electricity cooling meat	kWh	2.48 × 10^6^	2.48 × 10^6^	2.48 × 10^6^	2.48 × 10^6^	2.62 × 10^5^	4.78 × 10^6^
Total electricity	kWh	1.52 × 10^10^	3.33 × 10^9^	3.33 × 10^9^	9.89 × 10^7^	2.02 × 10^9^	5.37 × 10^9^
Electricity cost	USD	6.73 × 10^8^	1.47 × 10^8^	1.47 × 10^8^	4.36 × 10^6^	8.94 × 10^7^	2.37 × 10^8^
Volume water used by facility	m^3^	1.40 × 10^8^	3.06 × 10^7^	3.06 × 10^7^	8.72 × 10^5^	1.75 × 10^7^	5.64 × 10^7^
Annual water cost	USD	2.40 × 10^8^	5.23 × 10^7^	5.23 × 10^7^	1.49 × 10^6^	3.00 × 10^7^	9.65 × 10^7^

**Table 6 foods-11-03037-t006:** Outcome values relating to labor produced by the model for four mammalian scenarios proposed in the Risner et al. paper compared to outcomes for Hi-Five/Sf-9 and S2 insect cells [15].

		Mammalian [15]				Insect	
	Units	Scenario 1	Scenario 2	Scenario 3	Scenario 4	Hi-Five/Sf-9	S2
Annual manpower cost	USD	5.19 × 10^3^	3.54 × 10^2^	3.54 × 10^2^	5.10 × 10^1^	4.72 × 10^3^	1.58 × 10^4^
Annual labor cost	USD	1.57 × 10^9^	1.07 × 10^8^	1.07 × 10^8^	1.54 × 10^7^	1.43 × 10^9^	4.76 × 10^9^

**Table 7 foods-11-03037-t007:** Outcome values relating to finances produced by the model for four mammalian scenarios proposed in the Risner et al. paper compared to outcomes for Hi-Five/Sf-9 and S2 insect cells [15].

		Mammalian [15]				Insect	
	Units	Scenario 1	Scenario 2	Scenario 3	Scenario 4	Hi-Five/Sf-9	S2
Min. meat production to meet expenditures	kg	4.37 × 10^5^	5.73 × 10^4^	4.46 × 10^4^	1.96	4.20 × 10^3^	6.42 × 10^3^
Min. total annual expenditure	USD	5.29 × 10^13^	6.93 × 10^12^	5.40 × 10^12^	2.39 × 10^8^	5.07 × 10^11^	7.78 × 10^11^
Min. price of meat per kg	USD	USD 437,205	USD 57,291	USD 44,609	USD 2	USD 4193	USD 6426

**Table 8 foods-11-03037-t008:** Results of Sobol Sensitivity Analysis. AA2P, NaHCO_3_, and insulin concentration were omitted from table due to first order and total values equal to zero.

Variable	1st Order	Total
Average cell volume	2.29 × 10^−1^	8.38 × 10^−1^
Glucose conc. in basal media	7.30 × 10^−2^	7.90 × 10^−1^
Base media cost	7.61 × 10^−3^	1.34 × 10^−1^
Glucose consumption rate	3.68 × 10^−3^	1.27 × 10^−1^
Doubling time	4.02 × 10^−4^	9.33 × 10^−4^
Achievable cell concentration	−3.36 × 10^−5^	1.08 × 10^−5^
Oxygen consumption rate	−2.16 × 10^−8^	5.05 × 10^−11^
Specific heat of meat	1.80 × 10^−12^	6.54 × 10^−20^

**Table 9 foods-11-03037-t009:** List of cell types with range of reported cell diameters from the literature, average cell volumes, and corresponding prices of meat per kilogram.

Cell Type	Diameter (µm)	Average Volume (m^3^)	Price of Meat (per kg)
S2	10–12	5.73 × 10^−16^	USD 6425
High-Five	15–16.3	2.02 × 10^−15^	USD 4484
Sf-9	13–18.5	2.30 × 10^−15^	USD 3939
AeC6	30	1.14 × 10^−14^	USD 798

## Data Availability

All code for data analysis associated with the current submission is available at https://github.com/spanglab/ACBM_Calculator (accessed on 1 June 2021).

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
