# Peer review of "Entomoculture: A Preliminary Techno-Economic Assessment"

_foods, 2022, doi:10.3390/foods11193037_

Round 1
Reviewer 1 Report
The manuscript by Ashizawa et al. on Entomoculture: a preliminary TEA presents a very interesting study on the potential to optimize costs of cultivated meat production via replacement of mammalian cells by insect cells.
The way the authors adapted the open-source ACMB model to work with insect cell culture parameters is commendable, particularly concerning media replacement requirements (e.g. using lactate or ammonia accumulation as a determining factor).
However, I believe it is still needed to include more explanations on what the hypothetical product yielded by the insect cell cultivation would be. In the current manuscript version it appears to be taken for granted that simple biomass production using insect cells is sufficient to yield cultivated meat that can be observed as the traditional meat replacement. Still, I would expect that insect-cell-produced biomass would require significant post-processing to get to the stage where it resembles traditional meat in both texture and taste. This should be included in the discussion at least, if not in the model itself. Furthermore, it is not mentioned anywhere if the authors refer to only minced meat-like product as the end result of the insect cell cultivation? What about 3D steak-like products? If they are considered as well, then the use of scaffolds and associated costs of biomaterials, fabrication methods and other types of bioreactors should be counted in. In addition, even when considering minced meat-like only, it is not mentioned whether insect cell bioreactor cultivation requires microcarriers - I presume not, so that should also bring the costs down when compared to mammalian...in any case, it should be described if any support structures are intended for use for insect-based cultivated meat.
Another point which I find somewhat surprising is that the Antheraea clone “AeC6" were included only in the section when discussing the cell dimensions. Seeing that their use appears to significantly bring the total costs down, one needs to ask why are they not included in the full analyses in the same way as Hi-Five/Sf9 or S2 cells?
There are very few spelling mistakes:
line 9 - nutritious instead of nutritous
line 95 - presents instead of present
Based on all above, minor revisions are suggested for the manuscript to address listed remarks.
Author Response
Reviewer #1 comments:
However, I believe it is still needed to include more explanations on what the hypothetical product yielded by the insect cell cultivation would be, would require significant post-processing to get to the stage where it resembles traditional meat in both texture and taste. |
Added clarification of products to methods section, line 117-118. See below. Added reference to this as a limitation in the conclusion, lines 461-464. “Further research into this topic may explore costs associated with scaffolding or downstream processing associated with achieving more conventional-meat-like products (i.e., 3-D structure, texture, flavoring, etc.).”
|
It is not mentioned anywhere if the authors refer to only minced meat-like product as the end result of the insect cell cultivation? What about 3D steak-like products? If they are considered as well, then the use of scaffolds and associated costs of biomaterials, fabrication methods and other types of bioreactors should be counted in. it should be described if any support structures are intended for use for insect-based cultivated meat |
Added justification for the omission of scaffolds and microcarriers to the methods section, lines 115-120. “The cost model should be interpreted as a preliminary TEA, as certain processes such as seed train, scaffolding, microcarriers, and bioreactor cleaning are omitted. Downstream and post processing are also not considered within the original or present models, so the hypothetical product systems are assumed to yield products similar to minced meat. While these are certainly important factors for consideration in the cost estimation of cultured meat, maintaining included or excluded processes in the model constant between studies allows for comparison between the two cell types, which was the objective of the present study.”
|
Another point which I find somewhat surprising is that the Antheraea clone “AeC6" were included only in the section when discussing the cell dimensions. Seeing that their use appears to significantly bring the total costs down, one needs to ask why are they not included in the full analyses in the same way as Hi-Five/Sf9 or S2 cells? |
Robust data on necessary parameters for the model were only available for Hi-Five, Sf9, and S2 cells. |
Spelling: line 9 - nutritious instead of nutritous line 95 - presents instead of present |
This has been fixed. |

Reviewer 2 Report
The paper analyses the cost savings by utilizing insect cells in the cultivation of protein-enriched foods.
This topic is very interesting and very important in these years. The aim is well defined but in chapter 2 it is not clear why the authors decided to use the open-source ACMB cost model developed by Risner et al. and why the choice of lepidopteran ovarian cells.
The quality of presentation is low. In order to do a techno-economic assessment (TEA) it is necessary to know better the modeling of an industrial process. For this reason, in chapter 3 it is necessary to illustrate better the industrial process in different scenarios. Only after this presentation, in order to read the results, the authors could codify (maybe in the same table) the costs of each component in different scenarios: bio-equipment and media, electricity, water, labor. In this presentation it is not so easy to read the results.
In the paper it is also necessary to share the results to the discussion. The part of discussion needs to be strengthened (also using the results obtained in other studies).
The conclusions need to be re-write: the conclusions are very similar to the results and there is no reflection concerning why this paper could be useful for industrial companies. It could be useful add some reflections about future research need linked to the topic and limitations of this study.
Author Response
Manuscript ID: foods-1890578
Type of manuscript: Article
Title: Entomoculture: A Preliminary Techno-Economic Assessment
Authors: Reina Ashizawa, Natalie Rubio, Sophie Letcher, Avery Parkinson,Victoria Dmitruczyk, David L Kaplan*
Submitted to section: Meat
Reviewer #2 comments:
it is not clear why the authors decided to use the open-source ACMB cost model developed by Risner et al.
|
This cost model was purposely made open-source for use by the public to compare cultured meat product systems. This is stated in lines 84-86. |
why the choice of lepidopteran ovarian cells |
Two lepidopteran cell lines and one Drosophila cell line were used due to data availability. Lines 124-129 were edited to convey the reasoning: “Three insect cell lines were the focus of this study— Sf-9, Hi-Five, and S2— chosen for their common use in the pharmaceutical industry and robust data availability. Both originate from lepidopteran ovarian cells, the Sf-9 and Hi-Five cell lines had similar characteristics and were therefore analyzed together. S2 cells, which were established from Drosophila melanogaster embryonic tissue and are the most used Drosophila cell line, were analyzed independently due to their unique characteristics.” |
In order to read the results, the authors could codify (maybe in the same table) the costs of each component in different scenarios: bio-equipment and media, electricity, water, labor. |
Table 2 was added to directly compare costs of components. Lines 180-185 were added to more broadly discuss results before deeper analysis. |
the conclusions are very similar to the results and there is no reflection concerning why this paper could be useful for industrial companies. It could be useful add some reflections about future research need linked to the topic and limitations of this study. |
Section 5 has been edited to address this, and reflections on limitations and future directions have been added. |

Reviewer 3 Report
The manuscript "Entomoculture: A Preliminary Techno-Economic Assessment" reports a thorough analysis of different cell line cultures. The comparison in particular is between insect cell lines and mammalian cell lines. The topic is of interest as both, insects and cell culture represent potential food/protein sources. The manuscript contains a lot of data and details in particular about costs of production under different scenarios. The use of $ as metrics is useful also to made relative comparison between cell lines, however it should be stressed which factors mostly effect the costs (energy, raw materials) and discuss how this potential food source could be impacted by global crisis.
In the results and discussion section the "understanding model outcomes" paragraphs 3.1 to 3.6 describe methods used to define models and to analyze data. These should be moved to materials and methods section.
Line 87: paragraph number 1.3 was already used in the previous one.
Line 118: 2 cell lines?
Lines 413-424: here you state that the production of cultivated meat should avoid serum of animal origins and in particular FBS. However previously (line 99) you reported that the possibility to grow in serum free media was an advantage for insect cell lines. Please explain and check the text for consistency about this point.
Author Response
Manuscript ID: foods-1890578
Type of manuscript: Article
Title: Entomoculture: A Preliminary Techno-Economic Assessment
Authors: Reina Ashizawa, Natalie Rubio, Sophie Letcher, Avery Parkinson,Victoria Dmitruczyk, David L Kaplan*
Submitted to section: Meat
Reviewer #3 comments:
It should be stressed which factors mostly effect the costs (energy, raw materials) |
Added lines 180-186 and more text about this in the conclusion. |
Results and discussion section the "understanding model outcomes" paragraphs 3.1 to 3.6 describe methods used to define models and to analyze data. These should be moved to materials and methods section. |
Section 2.1 describes methods used to define insect models, sections 3.1-3.6 discuss the major outcome differences between scenarios while tracking and highlighting differences in the model to explain these differences |
Paragraph number 1.3 was already used in the previous one. |
This has been fixed. |
2 cell lines? |
Changed to two cell types (insect and mammalian) |
Here you state that the production of cultivated meat should avoid serum of animal origins and in particular FBS. However previously (line 99) you reported that the possibility to grow in serum free media was an advantage for insect cell lines. Please explain and check the text for consistency about this point. |
Lines 417-426 reworded for clarity. “Oftentimes, cell cultures will be supplemented with fetal bovine serum (FBS), but this is a byproduct of the meat industry, while in contrast, this model assumes the use of recombinant growth factors as animal serum alternatives. Cell cultures for cultivated meat can instead utilize these animal-free growth factors to increase proliferation, but as seen in the original model, these impose some of the most significant costs to production. As emphasized earlier in the text, insect cells can grow in the absence of these costly growth factors included in the original mammalian cell-based model, such as transforming growth factor beta and fibroblast growth factor 2. As increased cell concentration was found to be a considerable cost lever, other growth factors specific to insect cells have been identified and may be added to promote growth while not significantly contributing to cost of production.” |

Round 2
Reviewer 2 Report
no comments